# Water Surface Targets Detection Based on the Fusion of Vision and LiDAR

**DOI:** 10.3390/s23041768

**Published:** 2023-02-04

**Authors:** Lin Wang, Yufeng Xiao, Baorui Zhang, Ran Liu, Bin Zhao

**Affiliations:** 1School of Information Engineering, Southwest University of Science and Technology, Mianyang 621010, China; 2Laboratory of Science and Technology on Marine Navigation and Control, China State Shipbuilding Corporation, Tianjin 300131, China; 3Engineering Product Development Pillar, Singapore University of Technology and Design, Singapore 487372, Singapore; 4Tianjin Navigation Instrument Research Institute, Tianjin 300131, China

**Keywords:** target detection, CornerNet-Lite network, data fusion, bounding box

## Abstract

The use of vision for the recognition of water targets is easily influenced by reflections and ripples, resulting in misidentification. This paper proposed a detection method based on the fusion of 3D point clouds and visual information to detect and locate water surface targets. The point clouds help to reduce the impact of ripples and reflections, and the recognition accuracy is enhanced by visual information. This method consists of three steps: Firstly, the water surface target is detected using the CornerNet-Lite network, and then the candidate target box and camera detection confidence are determined. Secondly, the 3D point cloud is projected onto the two-dimensional pixel plane, and the confidence of LiDAR detection is calculated based on the ratio between the projected area of the point clouds and the pixel area of the bounding box. The target confidence is calculated with the camera detection and LiDAR detection confidence, and the water surface target is determined by combining the detection thresholds. Finally, the bounding box is used to determine the 3D point clouds of the target and estimate its 3D coordinates. The experiment results showed this method reduced the misidentification rate and had 15.5% higher accuracy compared with traditional CornerNet-Lite network. By combining the depth information from LiDAR, the position of the target relative to the detection coordinate system origin could be accurately estimated.

## 1. Introduction

With the increase of military and civil affairs applications of USV (Unmanned Surface Vessel), it is becoming important to enhance the USV’s ability to identify and locate water surface targets. In order to ensure maritime safety, the ability to identify and locate warship targets is very important to protect maritime territorial rights and interests [1]. Additionally, in modern marine operations, precise weapons guidance has high requirements for identifying and localizing the target. In the civil sector, the USV for water surface cleaning and environment monitoring [2] also requires accurate target identification and localization.

In the present day, vision detection [3], LiDAR detection [4], and multi-sensor fusion detection [5] are the main methods used for target detection on the water surface. The mainstay of vision detection is the convolutional neural network that is used to train the target sample to obtain the model and feature information that can be used for subsequent detection. However, it is easily affected by factors such as light, foam, reflection, and others present in a water environment. A LiDAR sensor learns through multi-view input, which can estimate the three-dimensional geometric structure of a target. While 3D LiDAR has been widely used in target detection, there are still many difficulties associated with its practical application. The first is that target segmentation is not easy in the background environment, especially for dynamic targets. Secondly, most current methods are based on surface texture or structural characteristics, which depend on the amount of data. Finally, the computation efficiency becomes very low when it comes to meeting the accuracy requirement.

The fusion of multi-sensor data provides multi-level information from different sensors combination, resulting in a consistent interpretation of the observation environment. The actual scene presents two challenges. The view angle of the sensor is the first one. It is important to understand that the camera images are obtained from the cone of vision, while the LiDAR point clouds are acquired from the real world. The second difficulty is the different data representations. Since the image information is dense and regular, while the point cloud information is sparse and disordered, the fusion in the feature layer will have an impact on the computation effectiveness due to different representations. CornerNet-Lite network is a lightweight network of target detection. Compared with the YOLO series algorithms, the methods based on this network is improved in terms of accuracy and speed. In such structure, different sizes targets are scaled, and small objects on water surface are easier to be found. Next, CornerNet-Lite network is adopted in this paper.

We propose a detection method based on the fusion of 3D LiDAR point clouds and visual information to identify and locate water surface targets. We achieve the target detection with the following three steps. Firstly, the model is trained using the CornerNet-Lite [6] lightweight network, and the confidence and pixel coordinates of the target are calculated. Simultaneous determination of visual inspection bounding boxes. Secondly, the processed point clouds are projected onto the two-dimensional pixel plane according to the extrinsic calibration matrix of the camera and LiDAR, and the pixel area of the point clouds is determined with the projection point cloud amount in the bounding box from the previous step. The LiDAR detection confidence (LDC) is determined by the bounding box area and the point cloud pixel area, the target confidence (TC) is determined by combining the camera detection confidence (CDC) with the LiDAR detection confidence (LDC), and the water surface target is determined by combining the detection threshold with the TC. Finally, the bounding box of the detection result is used to obtain the corresponding 3D point clouds. The relative position between the target and the detection coordinate system origin (DCSO) is obtained by calculating the 3D minimum bounding box of the point clouds. Experiments demonstrate that this method eliminates the influence of ripples and reflections in the water and reduces the false rate from water target recognition. With the LiDAR depth information, the target position in the detection coordinate system can be accurately computed.

The remainder of the paper is organized as follows: a discussion of some research explanations related to this topic is provided in Section 2. Detailed explanations of the proposed approach are provided in Section 3. The results and analysis of the experiment are presented in Section 4. Conclusions are drawn in Section 5 with a discussion of future works.

## 2. Related Work

### 2.1. Water Surface Targets Detection Based on the Vision

For the traditional water surface target detection method, researchers made use of a large number of sliding windows with different sizes to traverse each image from the camera, and then use the manual feature [7,8] and support vector machine [9] classifier to recognize the target object. As the sliding process of windows is redundant and the expression capability of manual features is limited, it is very difficult to adapt traditional methods to detect targets on water surfaces. Recent advances in convolutional neural networks have made this detection possible. The methods from such advances are typically divided into two types: one-stage methods and two-stage methods. The two-stage method first extracts the candidate areas from the image and then classifies the targets within the candidate areas and regresses the bounding box, such as Fast R-CNN [10] and Faster R-CNN [11]. The one-stage approach estimates the target directly based on a pre-defined anchor frame, such as YOLO [12,13,14] or SSD [15]. The two-stage method has a greater advantage in terms of detection accuracy, while the one-stage method has higher efficiency.

Lin [16] et al. introduced the channel attention mechanism into Faster R-CNN, which could improve the target detection accuracy by suppressing redundant features. Cheng [17] et al. used blank label training and optimized YOLOv4 to augment the target network. Ma [18] et al. improved the YOLO v3 and KCF algorithms to obtain accurate identification and real-time tracking of multiple targets on the water surface. For the particularity and complexity of the water surface environment, it is generally recommended to improve the accuracy of the detection accuracy by improving the training network model, but the impact of reflection and illumination has not been adequately addressed.

### 2.2. Water Surface Targets Detection Based on LiDAR

In addition to restore the target’ three-dimensional dynamic information, such as the shape, size, and space, the 3D LiDAR can capture the target point cloud in real-time. According to the input format, these methods fall into three general categories: point clouds, images, and voxels. For the method of directly operating point cloud data, Charles [19] and others put forward an improved network called PointNet++. It is capable of learning data with different scales. Concerning the method of inputting the image format after point cloud projection, Chen et al. [20] employed 2D point cloud detection networks as the framework to fuse the vision information, thereby enriching the features. As a method for voxel-based input, Muratura et al. [21] proposed the VoxNet network in 2015. In this study, LiDAR point cloud data are voxelized and combined with 3D convolution to use point cloud for network training, thereby presenting a new method for target recognition. Zhou et al. [22] used a voxel network to predict the 3D bounding box of LiDAR points.

It has been verified and analyzed by Stateczny et al. [23] that water targets can be detected by a mobile 3D LiDAR, and several small targets can be detected except inflatable targets (such as fenders or air toys). Based on 3D LiDAR, Zhou et al. [24] presented a joint detection algorithm for water surface targets called DBSCAN-VoxelNet. This algorithm has an excellent performance in suppressing clutter on the water surface. Ye et al. [25] clustered the LiDAR point clouds by improving the DBSCAN algorithm, which both recognizes close obstacles and targets at tens of meters away. Zhang et al. [26] proposed a method to detect mobile targets in water based on LiDAR point clouds and image features. The combination of camera image and LiDAR point were used to determine the types of ground targets or obstacles, the detection confidence, the distance, and the azimuth information of unmanned surface vehicles. In order to perceive, detect, and avoid obstacles, Chen et al. [27] incorporated multidimensional environmental information from camera and LiDAR. The sensor network adopted in literature [28] is also an alternative to replace our communication system. Although the point cloud information can be enriched by clustering, it is difficult to identify objects accurately, for the insufficient features from the point cloud.

## 3. Proposed Approach for Target Detection Based on LiDAR and Vision Fusion

### 3.1. Overview of the Proposed System

Using the camera solely to detect targets on the water surface may result in misidentification, identification loss, etc, due to factors such as target reflection, ripples on the water surface, and reflections on the water surface. Thus, this paper proposes a method for detecting targets on the water surface by combining LiDAR and a camera. The entire process involves three stages. In the first stage, the LiDAR and camera are used to collect the original data, and the image information is then trained by CornerNet-Lite in order to achieve target classification and confidence estimation. In the second stage, a three-dimensional point cloud is projected onto a two-dimensional pixel plane through the joint calibration of the camera and LiDAR, and a point cloud’s confidence level is determined by the ratio between the pixel area occupied by the point clouds and the one of the bounding boxes. The experiments were performed to obtain an optimal weight ratio, according to the weight ratio fusion camera and LiDAR information, and then to achieve accurate classification and detection of water targets, to obtain the fusion detection of the bounding box. In the third stage, the target position is determined using the bounding box, which is framed with the minimum bounding box of the AABB(Axis-aligned bounding box). The positioning result is defined as the central coordinate of the bounding box. As shown in Figure 1, the system structure is depicted. These three steps correspond to the following three subsections.

### 3.2. Detection of the Water Surface Target Based on CornerNet-Lite

The vision detection part employs the CornerNet-Lite network. The structure can be found in Figure 2. Initially, the raw image collected by the camera is input into the detection network, and the raw image data are compressed into 255 × 255 and 192 × 192 data, and the latter is expanded to 255 × 255. According to the pixel size, the objects are defined as the small target, the medium one, and the large one. With the attention maps corresponding to the possible locations, sizes, and scores of targets’ presence, these targets are predicted based on different features. In order to find the most possible target areas, the first k areas with scores exceeding a specified threshold are selected as possible target areas, and these locations are input into the lightweight Hourglass network. Based on the target frame size in the results, the detection area is enlarged once to prevent the small targets from being missed. Finally, a non-maximum suppression algorithm is used to optimize the detection of edges.

As a one-stage detection algorithm, a large number of anchor frames are not used in the CornerNet-Lite network. Among the trained image samples, only the position recognized as ground-truth is considered a positive sample, whereas the other positions are considered a negative sample. During the training process, only the negative sample loss computation in a certain circular domain near the positive sample position is reduced. The reason is that when the position close to the positive sample is incorrectly detected as the key corner, the false detection result will still generate the target bounding box and overlap the bounding box that has been correctly detected. Consequently, the circle radius in this area should satisfy the requirement that the intersection ratio of positive samples within this radius range is greater than 0.7, and the reduction in loss calculation satisfies 2D Gaussian distribution $e^{-\frac{x^{2}+y^{2}}{2\delta^{2}}}$ $\left(x^{2}+y^{2}\leq \delta^{2}\right)$, where x and y represent the distances from the pixel point to the active sample, while 2$\delta^{2}$ represents the variance, i.e.,
(1)$$y_{cij}=\left\{ \begin{array}{l} e^{-\frac{x^{2}+y^{2}}{2\delta^{2}}(x^{2}+y^{2}+\delta^{2})}\\ 0 \end{array}\right.$$

If our intersection ratio is less than 0.7, there will be a large case of false recognition and reduce the accuracy of the system. Based on Equation (1), the optimized focus loss function is as follows:(2)$$L_{\det}=\frac{-1}{n}\sum_{c=1}^{C}\sum_{x=1}^{H}\sum_{y=1}^{W}\left\{ \begin{array}{l} {\left(1-P_{cij}\right)}^{\alpha }\log(P_{cij})\\ {\left(1-y_{cij}\right)}^{\beta }{\left(P_{cij}\right)}^{\alpha }\log(1-P_{cij}) \end{array}\right.$$ where *n* represents the total number of detected targets in the image. When $y_{cij}$ = 1, the formula in the upper bracket is satisfied, and the super-parameters are controlled by the variables α and β.

In the CornerNet-Lite backbone network, Hourglass performs up-and-down sampling from the input image sequence. Simply up-sampling and down-sampling will cause a reduction in image size, which will cause a loss of detection accuracy. Therefore, smoothing loss is used to resolve this issue. The key idea is to perform a rounding operation on the reduced coordinates during down-sampling, the reduced coordinates are rounded. For one point (x,y) in the image, down-sampling i times maps correspond to the coordinate $\left(\left[\frac{x}{i}\right],\left[\frac{y}{i}\right]\right)$ on the thermal map, and to restore the image size, there will be a coordinate offset after up-sampling. The offsets are expressed as follows:(3)$$\Delta_{k}=\left\{\frac{x_{k}}{i}-\left[\frac{y_{k}}{i}\right],\frac{x_{k}}{i}-\left[\frac{y_{k}}{i}\right]\right\}$$ where *k* represents the number of target targets in the image, and $\left(x_{k},y_{k}\right)$ represents the coordinate of the *k-th* target. A smooth *L*1 loss is adopted at the corner points of the positive sample when the network makes a prediction as follows.
(4)$$Foff=\frac{1}{n}\sum_{k=1}^{n}smoothL1Loss(O_{k},\hat{O_{k}})$$

Although the CornerNet-Lite network can identify most targets, it can be influenced by spray, reflection, underwater disturbances, and ripple, leading to target misidentifications, because of the instability of the water surface.

### 3.3. Camera and LiDAR Data Fusion for Target Detection

For camera and LiDAR, external parameter calibration is required prior to sensor data fusion. Each small square on the calibration board has a side with 108 mm length, and it is arranged as a chessboard with 8 rows and 6 columns. During calibration, LiDAR and camera data from the calibration plane are collected. An illustration of the joint calibration model is shown in Figure 3.

A $T_{cl}$ matrix represents the conversion matrix from a camera coordinate system to a LiDAR coordinate system, and a $T_{cm}$ matrix represents a conversion matrix from a calibration board coordinate system to a camera coordinate system. During the calibration process, the Calibration Toolkit module of Autoware is used. As a result of the calibration, the calibration result is automatically calculated, and the extrinsic calibration matrix of the camera and LiDAR is as follows:$$T_{c1}=\left[\begin{array}{cc} R_{c1} & t_{c1}\\ 0^{T} & 1 \end{array}\right]=\left[\begin{array}{cccc} -0.0501 & 0.152 & 0.987 & 0.119\\ -0.999 & -0.00815 & -0.0494 & 0.0193\\ 0.000548 & -0.988 & 0.152 & -0.0492\\ 0 & 0 & 0 & 1 \end{array}\right]$$
where $R_{cl}$ matrix is a 3 × 3 matrix representing the rotation of the camera with respect to the LIDAR, and $t_{c1}$ is a 3 × 1 translation matrix representing the translation of the camera with respect to the LiDAR. The matrix values are obtained using Autoware’s Calibration Toolkit module, and the result represents the rotation translation of the camera in relation to the LiDAR. The LiDAR data can be transformed into the camera coordinate system using this transformation matrix. To achieve the data fusion from camera and LiDAR, the 3D point cloud data are projected to the pixel plane with the internal reference matrix obtained from the camera calibration.

The 3D point cloud data are projected into the image using the Formula (5), which $\left(X_{t},Y_{t},Z_{t}\right)$ represents the position of the target in the camera coordinate system, (u,v) represents the pixel coordinates, and $Z_{c}$ represents the camera internal reference matrix. We collect data with the camera and LiDAR simultaneously and project the 3D point clouds onto the camera plane with the $T_{cl}$ obtained earlier. The projection effect is shown in Figure 4.
(5)$$Z_{c}\left[\begin{array}{c} u\\ v\\ 1 \end{array}\right]=K=\left[\begin{array}{cc} R_{c1} & t_{c1} \end{array}\right]\left[\begin{array}{c} X_{t}\\ Y_{t}\\ Z_{t}\\ 1 \end{array}\right]$$

Considering that the LiDAR scans 360 degrees in all directions, while the camera has a relatively fixed view field, accurate projection results can be obtained when the target lies within the co-view area of the camera and the LiDAR. Figure 4 gives an example of the LiDAR projection. Figure 4a represents the raw image of the target as seen by the camera view field. Figure 4b shows the raw LiDAR point clouds of a target in camera view, where it can be seen that water is transparent to laser rays. Figure 4c is the projection of the 3D point cloud data of the target into a 2D pixel coordinate system.

When performing water surface target detection, it is prone to misidentification due to the influence of light, ripples, and reflections. After the introduction of LiDAR point cloud data, the number of point clouds in each recognition box is counted to determine whether the target box is a real target, thus playing a role in eliminating false recognition. The algorithm for fusion camera and 3D LiDAR detection is shown in Algorithm 1.
**Algorithm 1:** Fusion detection of LiDAR and cameraInput:   Input a picture and the corresponding PCDOutput:   Fusion detection results   Target box diagonal coordinates 1: Get the result of CornerNet-Lite 2: Have Sbox,Nbox,Confcamera3: For all i in Sbox do4:   For all point 3d in point 2d, do5:     Get **I** (x, y)6:   End for7:   Statistical points quantity Slaser in Sbox8:   Conflaser=λ∗(Slaser/Sbox)9:   Fusion confidence: α1Conflaser+α2Confcamera>Y10: End for

With CornerNet-Lite, most floating targets on the water surface can be detected and located, with the target coordinates and the detection confidence degree of each target, which we define as the CDC degree $Conf_{\mathrm{camera}}$. After the 3D point clouds data have been obtained from the LiDAR, they are converted into a PCD format with a system time stamp, while making sure that each PCD frame corresponds to its image frame. According to the conversion matrix obtained from the camera calibration, the 3D point cloud is projected onto the 2D image plane. Considering a large number of 3D point clouds, it is necessary to filter some with their coordinate values before projection, thus reducing the amount of computation required. On the 2D image plane, the set of projected points I have the coordinate (x,y) , and the principle of the screening is shown in Equation (6).
(6)$$I=\left\{ \begin{array}{l} I\cup (x,y)\;if(0\leq x\leq 640)\;and\;(0\leq y\leq 480)\\ I\cup \emptyset \;\;otherwise \end{array}\right.$$

For the sample image of the water environment, after projecting the point clouds onto the 2D image plane, the water environment does not have any projected point clouds, while the projected point clouds will be distributed on the floating target. We define the ratio of the projected area of the point clouds to the area of the bounding box pixels as the LDC $Conf_{laser}$, as illustrated in Equation (7).
(7)$$Conf_{laser}=\left\{ \begin{array}{l} \lambda *\frac{S_{laser}}{S_{bbox}}\;\lambda *\frac{S_{laser}}{S_{bbox}}<1\\ 1\;\;\;\mathrm{otherwise} \end{array}\right.$$

As shown in the equation above, $S_{bbox}$ represents the pixel area of the prediction box corresponding to the candidate target, $S_{laser}$ represents the pixel area occupied by the point clouds in the prediction box in the fused image, and λ represents the pixel area adjustment factor, that is, the proportion of the pixel area that corresponds to the adjusted point clouds data in the pixel area of the prediction box. According to 400 statistics test sets, the number of point clouds reflected back from a little object was small, with a ratio of point cloud area to prediction boxes of approximately 0.015 to 0.020. For small to medium-sized floating targets, the ratio between the area of the point clouds and the area of the prediction box is 0.020 to 0.025. In addition, the larger floats have an area ratio at least 0.025. This ratio occurs most frequently around 0.025, as shown in Figure 5. Therefore, the pixel area adjustment factor λ in this paper is taken as 40. All results are taken as 1 when $\lambda *(S_{laser}/S_{bbox})$ are greater than or equal to 1. In the case of area ratio for small target detection, $Conf_{laser}$ takes the middle value of 0.7.

Following data fusion, the final confidence determination is shown in Equation (8). $Conf_{cammera}$ represents the confidence level obtained from the detection of the target by the CornerNet-Lite network. $Conf_{laser}$ represents the confidence level of LiDAR detection as defined in Equation (7). $\alpha_{1}$, $\alpha_{2}$ represent the weighting coefficients and the sum of the two values is 1. The target is considered to be detected when the final obtained target confidence level Conf is greater than a set threshold *Y*. Otherwise, it is determined that no target is detected or that reflections or ripples cause interference. In visual inspection, we use a non-maximum suppression method to mark targets with $Conf_{camera}$ greater than 0.7. At the same time, we only keep the detection results with $Conf_{\mathrm{laser}}$ greater than 0.7. With linear combination, our fusion detection confidence level is greater than 0.7.
(8)$$Conf=\left\{ \begin{array}{l} \alpha_{1}Conf_{camera}+\alpha_{2}Conf_{laser}\\ \alpha_{1}+\alpha_{2}=1 \end{array}\right.$$

The statistical results shown in Figure 6 came from 200 frames of image data and LiDAR point cloud. In these data, there are over 1000 floating targets. The number of validly detected targets is determined by varying the image confidence weight *α*1 and the final decision threshold *Y*. According to the statistical results, with a determination threshold *Y* greater than 0.6, more floating targets are sifted out and an excessively high confidence level threshold is not appropriate. Even though the targets are essentially detected when the determination threshold is lower than 0.4, there will occur multiple frames of the same target, or even misidentification. Therefore, a too-low determination weight is also inappropriate. With a *Y* value of 0.5 and *α*1 value of 0.6, the best detection is achieved.

### 3.4. Creation of the 3D Minimum Wraparound Box

With a simple shape and slightly larger volume, a wraparound box involves wrapping a complex shape model in a bounding box. It is widely used in collision detection and ray tracing. After replacing the modeled target with a wraparound box, the light first makes a quick intersection test with the box. If the light does not cross the box, it does not intersect the modeled target. Since elements that do not intersect are eliminated, the intersection tests number is significantly reduced, and geometric computations become more efficient. Considering the relationship among the wraparound box tightness, the intersection test complexity and the boxes number, this paper adopts the AABB wraparound box with a simple intersection and poor tightness. According to the AABB definition, it is the smallest 6 hexahedron that wraps a target, and its edges are parallel to the axes. Therefore, an AABB can be described with only six parameters.

Make the flush coordinates of camera image pixel as $p_{c}=(u_{c},v_{c},1)$, the coordinates under the camera coordinate system {C} as $P_{c}={\left(X_{c},Y_{c},Z_{c}\right)}^{T}$, the coordinates under the LiDAR coordinate system {L} as $P_{l}={\left(X_{l},Y_{l},Z_{l}\right)}^{T}$, corresponding to the coordinates in the world coordinate system as $P_{w}={\left(X_{w},Y_{w},Z_{w}\right)}^{T}$. As a final localization result, the target position needs to be converted to the coordinates under the world coordinate system. In Figure 7, the conversion model is shown. From the pinhole camera model, we can obtain:
(9)$$Z_{c}p_{c}=KP_{c}$$

Suppose $R_{c}$ and $t_{c}$ are the rotation and translation matrices of {C} relative to {W} and point $P_{c}$ are related to point $P_{l}$ as follows:(10)$$P_{c}=R_{c}P_{w}+t_{c}$$

Suppose $R_{l}$ and $t_{l}$ are the rotation and translation matrices of {L} relative to {W} and point $P_{c}$ are related to point $P_{w}$ as follows:(11)$$P_{l}=R_{l}P_{w}+t_{l}$$

Combining the two equations, we can obtain:(12)$$P_{l}=R_{l}R_{c}{}^{-1}P_{c}+t_{l}-R_{l}R_{c}{}^{-1}t_{c}$$

Therefore, the rotation and translation matrices of the camera and the LIDAR are $R_{l}R_{c}{}^{-1}$ and $t_{l}-R_{l}R_{c}{}^{-1}t_{c}$, respectively, which are noted as $R_{cl}$ and $t_{cl}$. With joint calibration, these two parameters can be calculated. Combining Formula (9), we can obtain:(13)$$P_{l}=Z_{c}R_{cl}K_{c}{}^{-1}p_{c}+t_{cd}$$

Therefore, the coordinates $P_{w}$ of the point $P_{l}$ expressed under {W} are expressed as:(14)$$P_{w}=T_{lw}P_{l}=T_{lw}(Z_{c}R_{cl}K_{c}{}^{-1}p_{c}+t_{cd})$$

Inferring the 3D minimum wraparound box from the anchored box obtained from target detection, it is necessary to convert the 3D boundedness to a constraint on the 2D images by inferring the 3D minimum wraparound box from the anchored box obtained from target detection. Suppose the CornerNet-Lite detection result in image of the *i*-th image as $S_{i}$, and iterate over $S_{i}$ to obtain the top-left, top-right, bottom-left, and bottom-right vertex coordinates of the anchored frame as $T\left(u_{l},v_{l}\right)$, $T\left(u_{r},v_{r}\right)$, $D\left(u_{l},v_{l}\right)$, $D\left(u_{r},v_{r}\right)$, respectively. Based on Equation (14), the vertices coordinates can be determined. In this case, the four vertices are connected to the optical center of the monocular camera, resulting a quadrilateral cone with the camera’s optical center as its apex. Next, the target’s anchoring frame locates in the quadrilateral. Due to projecting and extending the quadrilateral from the optical center with several different viewpoints, the target 3D region can be obtained, and the maximum and minimum values on the three axes of the region are taken as the minimum bounding box in 3D, as illustrated in Figure 8.

Define a single 3D plane as $P^{i}=(p_{x}^{i},p_{y}^{i},p_{z}^{i},p_{w}^{i})$, the corresponding linear harness relationship can be listed as,
(15)$$P^{i}X\leq 0$$

AABB wraparound box can be expressed as:(16)$$R=\left\{(x,y,z)|\min_{x}\leq \max_{x},\min_{y}\leq \max_{y},\min_{z}\leq \max_{z}\right\}$$

In the formula, $\min_{x},\max_{x},\min_{y},\max_{y},\min_{z},\max_{z}$ represent the minimum and maximum values of the hotspot minimum box, which are projected on three rectangular axis. Each perspective corresponds to a quadrilateral which can be viewed as four planes intersecting. The six parameters of the AABB wraparound box is transformed into solving the intersection of 4n different planes. Therefore, in order to determine box parameter $\min_{x}$, the linear constraint relationship will be as follows:(17)$$p_{x}^{i}x+p_{y}^{i}y+p_{z}^{i}z+p_{w}^{i}\leq 0,i=\{1,2,\ldots ,4n\}$$

There is a similar calculation method for the other five parameters as well. The problem falls into the category of linear programming. By using the simple method, the optimal solution is obtained through finite-step iterations. An AABB wraparound box plot of the fused detection target in the camera view is shown in Figure 9.

In order to evaluate localization error, the Root Mean Square Error (RMSE) in [29] is introduced in this paper.

## 4. Experimental Results

We validated the proposed method with an USV as shown in Figure 10. The computer is equipped with an Intel i7-10750H processor running at 2.6 GHz, 16 GB of RAM and a GTX1650Ti GPU card, the operation system is Ubuntu 18.04 LTS, the LiDAR is a Velodyne VLP-16 3D, and the camera model is Spedal 920PRO. The relative position between the camera and LiDAR is shown in Figure 10.

### 4.1. Surface Target Detection

If only vision is used to detect targets on the water surface, excessive light and reflections can cause loss of recognition, and waves can cause misidentification. Since the laser ray emitted from LiDAR can pass through the transparent medium without returning the point clouds, this characteristic can be used to solve the above problems. All data for experiments were collected from one lake. This dataset contains 4800 640*480 pixels images, along with 2400 frames of LiDAR point cloud information. These 3D point cloud information are processed into PCD format aligned with the timestamp of the image. One image in 0.5 s is selected for detection with the PCD data.

As shown in Figure 11, CornerNet-Lite detected the following results. Figure 11a demonstrates the significant misidentification of vision detection as a result of ripple effects. Due to the reflection of the target in Figure 11b, a target appears to be misidentified as well. Similarly, Figure 11c illustrates misidentification due to the effects of light and perspective. Based on the results of this experiment, it is evident that tar-get identification in the water environment is affected by factors such as the reflection of the target and the ripples on the surface of the water. In order to address the above problem, we make use of 3D LiDAR to acquire the point clouds of the surface object on the water.

When the LiDAR data are merged with the camera data to examine the samples in Figure 12, it is clear that the misidentifications caused by ripples, reflections, lighting, and other factors are well eliminated, and this is because the LiDAR point clouds do not return data in the transparent medium. By recalculating the prediction frame confidence levels based on the fused confidence Equation (8), it is possible to significantly decrease the false identification rate. Due to its distance far from the LiDAR and the lack of LiDAR data on the target, the target in the top right corner of Figure 12b is not identified. This is due to the fact that our detection results by fusion are influenced by the LiDAR detection confidence parameter. After a large number of experiments, it is seemed that the detection results can achieve an optimal solution when the visual detection weight is 0.6 and the LiDAR weight results are at 0.4. These two weights can be modified accordingly for different experimental situations. From Equation (8), can be seen that the confidence level can reach above 0.7.

To verify the accuracy, our detection algorithm is compared with three algorithms: CornerNet-Lite, YOLOv3, and YOLOv5.For each algorithm, we count the number of correct recognitions and the number of false recognitions. The accuracy rate is calculated as the ratio of the number of correct recognitions to the total number of targets in the field of view, as shown in Table 1 and Figure 13.

Table 1 indicates that the recognition accuracy of YOLO v5 is relatively low because the water surface targets are small targets with small sizes, and the downsampling times of YOLO v5 is large, which makes it difficult to learn feature information about small targets. Although YOLO v3 has better detection results than YOLO v5, the detection accuracy of small targets remains relatively low, and some false recognitions occur. There was an accuracy improvement for CornerNet-Lite algorithm over YOLO v3, but the number of false positives increased to four. Based on CornerNet-Lite, the proposed algorithm fused LiDAR point clouds projection data. As a result, the false recognition is decreased and the accuracy is improved.

### 4.2. Positioning of the Target AABB Bounding Box

In Section 3.4, it is described about how the projected LiDAR point cloud is converted from camera pixel coordinates to world coordinates. The AABB bounding box for the target in Figure 9 corresponds to the fusion detection result in Figure 11c. With the standard distance measurement method, the location relative to the detection coordinate system origin is acquired, and the 3 coordinates values are defined as the “true position”. For the proposed algorithm, the center of the bounding box is defined as the “estimated position“. To validate the accuracy, the distance deviation between “true position” and “estimated position” is defined as an “error”. According to the “error”, it is obvious that the detection target can be localized relatively accurately. The positioning results are shown in Table 2.

Above objects’ position are based on the fused detection. To achieve three-dimensional positions of the targets, the two-dimensional pixel coordinates of the target are converted to the coordinates in detection system, with the same confidence level in Section 4.1. We calculated the RSME for x, y, z in this table: RSME(x) = 0.073, RSME(y) = 0.079, RSME(z) = 0.065. The RMSE are all less than 0.1, indicating that the positioning results are reliable.

## 5. Conclusions

This paper proposed a method for recognizing and localizing targets by fusing vision and LiDAR information for unmanned surface vessel. There are two important works implemented. Firstly, the boundary box of target is obtained by vision detection, and the influence of reflection and ripple is solved by fusing LiDAR data. Our algorithm has 15.5% improvement in accuracy compared with traditional CornerNet-Lite. Secondly, the pixel coordinates of the target are determined by fusing the detection results, and the target spatial coordinates in the detection coordinate system are calculated with AABB bounding box. Several experimental tests have demonstrated that the method is more accurate than CornerNet-Lite algorithm, and it can eliminate the misidentification caused by reflections and ripples on the water surface. With the 3D minimum bounding box from target image bounding box, the detection target position is calculated.

**Limitation and future work.** The obvious limitation of this paper is the resolution of a cheap mechanical LiDAR. The farther the target is from the LiDAR, the fewer point cloud data are acquired from the target surface. In the target localization, the main error reason is the time synchronization of the camera and LiDAR data. In future work, the solid-state LiDAR will be integrated with small target detection algorithms to enhance accuracy, and the inter-frame interpolation is used to achieve time synchronization between the camera and LiDAR to reduce localization errors. In addition, regarding the relation between visual detection weight and LiDAR weight, some principles deduction work will be arranged.

## Figures and Tables

**Figure 1 sensors-23-01768-f001:**
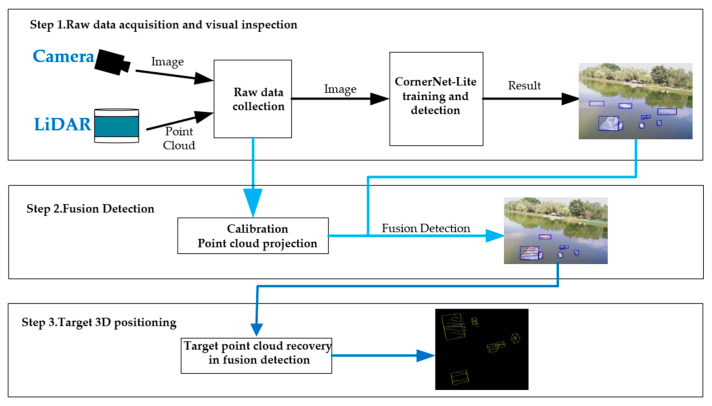
System Overview.

**Figure 2 sensors-23-01768-f002:**
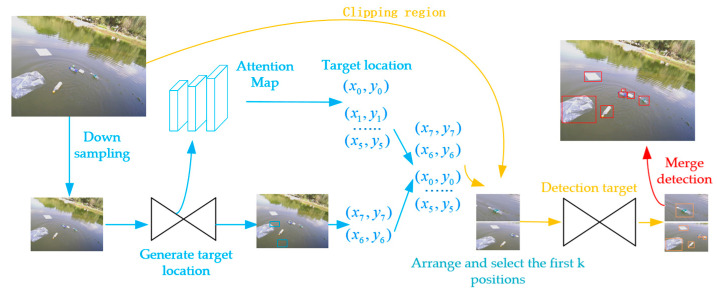
CornerNet-Lite Structure Diagram.

**Figure 3 sensors-23-01768-f003:**
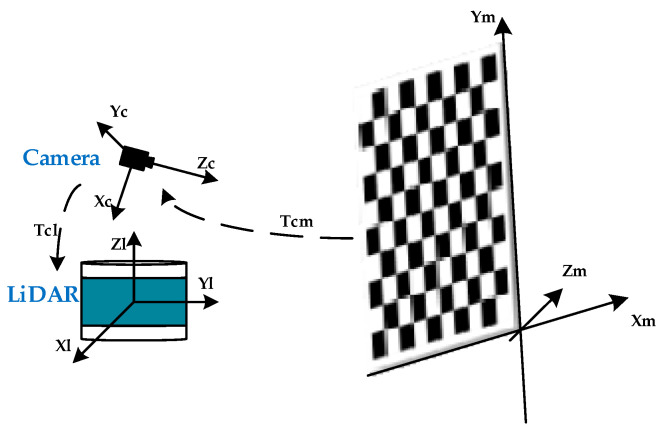
Joint Calibration Model.

**Figure 4 sensors-23-01768-f004:**
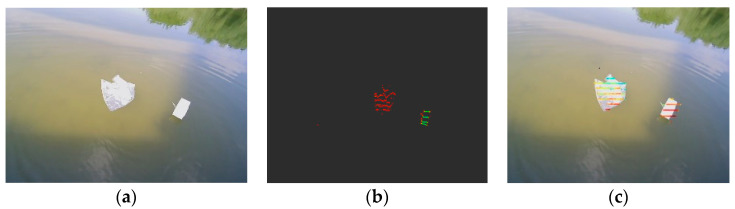
LiDAR point clouds projection. (**a**) Original picture of the camera. (**b**) Target point clouds projection. (**c**) Projection results.

**Figure 5 sensors-23-01768-f005:**
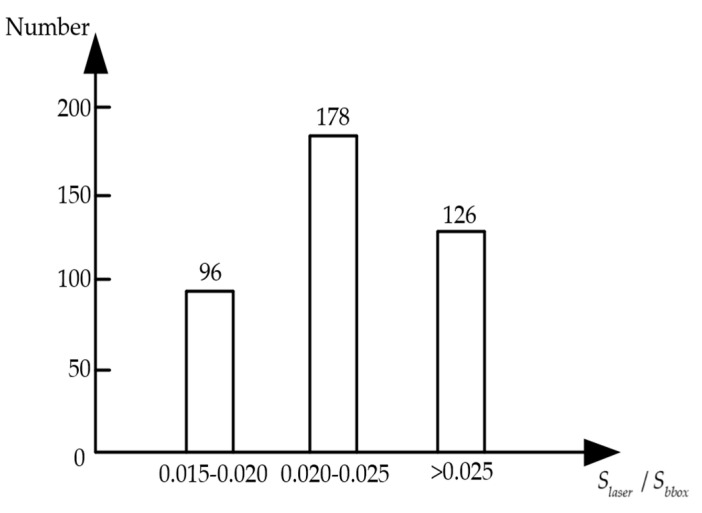
Histogram of point clouds area proportion statistics.

**Figure 6 sensors-23-01768-f006:**
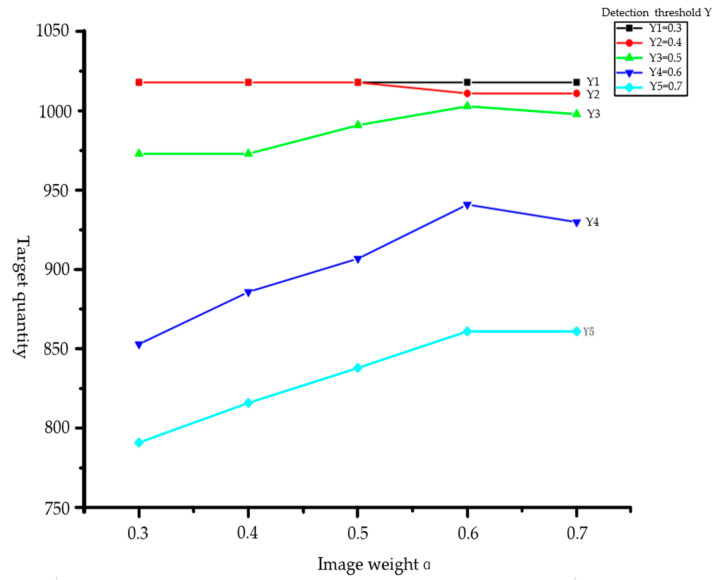
Threshold and weight values versus detection effect statistics.

**Figure 7 sensors-23-01768-f007:**
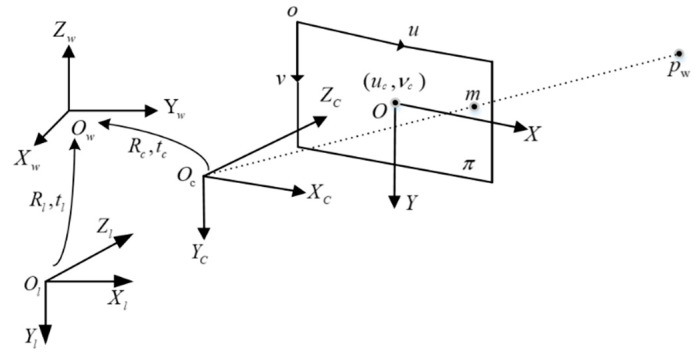
Conversion model.

**Figure 8 sensors-23-01768-f008:**
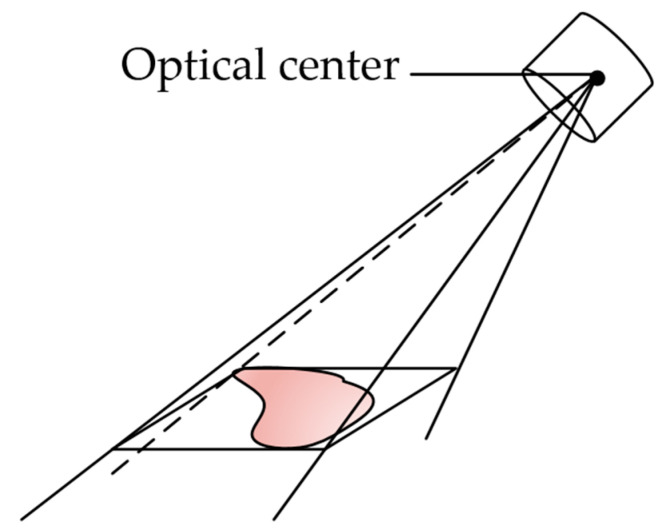
3D back-projection model of the anchored frame.

**Figure 9 sensors-23-01768-f009:**
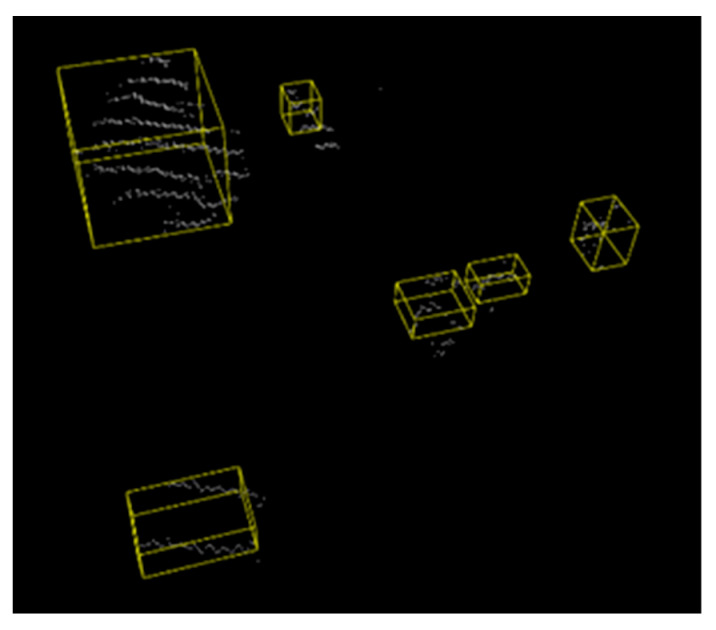
AABB wraparound box for fusion detection results.

**Figure 10 sensors-23-01768-f010:**
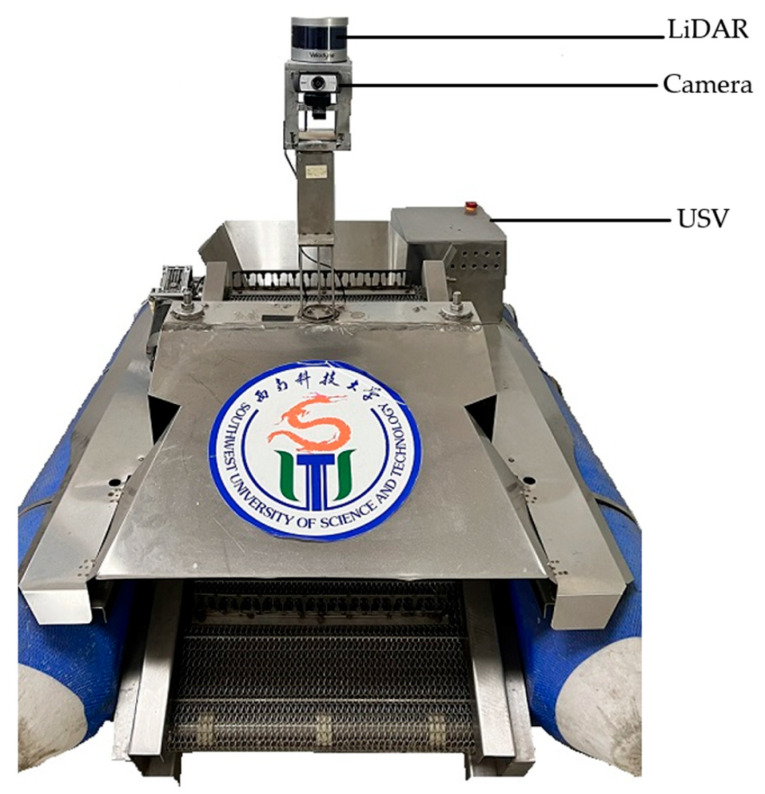
Hardware architecture diagram.

**Figure 11 sensors-23-01768-f011:**
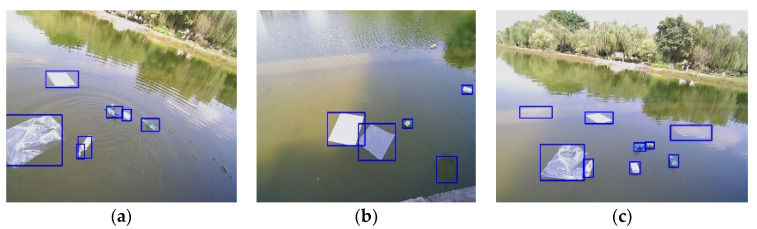
CornerNet-Lite detection results. (**a**) Light influence. (**b**) Water reflection. (**c**) Ripple effect.

**Figure 12 sensors-23-01768-f012:**
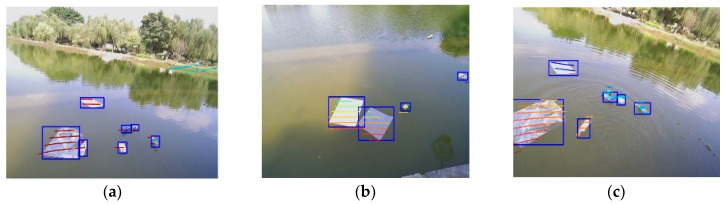
Fusion detection results. (**a**) Light influence. (**b**) Water reflection. (**c**) Ripple effect.

**Figure 13 sensors-23-01768-f013:**
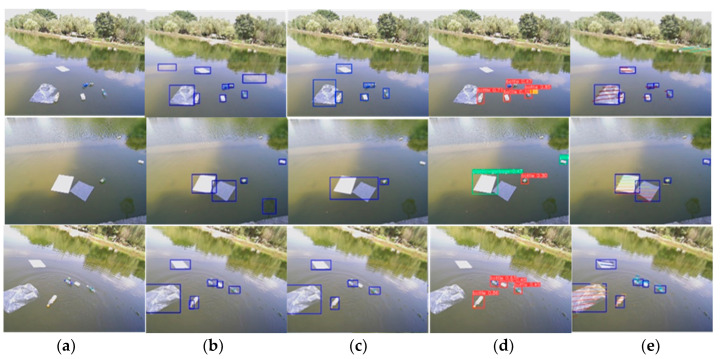
Comparison of the results of the four algorithms. (**a**) Original image. (**b**) CornerNet-Lite. (**c**) YOLO v3. (**d**) YOLO v5. (**e**) The algorithm in this paper.

**Table 1 sensors-23-01768-t001:** Recognition of the four algorithms in the three scenarios.

Algorithm	Number of Correct Identification	Number of Misidentifications	Accuracy/%
CornerNet-Lite	17	4	73.9
YOLO v3	13	1	65.0
YOLO v5	11	0	57.9
Our algorithm	17	0	89.4

**Table 2 sensors-23-01768-t002:** Target position estimates.

Target Serial No.	True 3D Position (m)	Estimated 3D Position (m)	Error (m)
Object 1	(2.01, −0.53, 0.02)	(1.964, −0.462, −0.04)	0.084
Object 2	(1.92, −0.11, 0.20)	(1.948, −0.037, 0.137)	0.100
Object 3	(1.52, 0.35, −0.31)	(1.395, 0.290, −0.230)	0.160
Object 4	(1.31, 0.88, −0.24)	(1.345, 0.761, −0.143)	0.157
Object 5	(2.22, 0.94, 0.63)	(2.196, 0.845, 0.585)	0.097
Object 6	(1.85, −0.12, 0.09)	(1.959, −0.147, 0.070)	0.198

## Data Availability

Not applicable.

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
