# Peer review of "Water Surface Targets Detection Based on the Fusion of Vision and LiDAR"

_sensors, 2023, doi:10.3390/s23041768_

Round 1
Reviewer 1 Report
This manuscript proposed a detection method based on the fusion of 3D point clouds and visual information to detect and locate water surface targets. It is an excellent research paper with complete content and rigorous structure. In addition, a complete experimental verification of the proposed method has been carried out.
But there are still a few questions. Please explain them clearly.
1. The first step of this method is to detect water surface targets using CornerNet-Lite network. However, it is not mentioned why to use CornerNet-Lite network. Please add a description in the manuscript.
2. Please explain the selection principle of 0.7 in line 184:” the circle radius in this area should satisfy the requirement that the intersection ratio of positive samples within this radius range is greater than 0.7”.
3. There is no explanation for the matrix value in line 223, please specify the reason for the matrix value.
4. The confidence level characteristics of the two systems are not analyzed, but the confidence levels of the two systems are directly combined linearly. Please explain why the confidence level can be combined linearly in line 283.
5. Please explain in detail the relationship between literature 1 :“Wearable Polarization Conversion Metasurface MIMO Antenna for Biomedical Applications in 5 GHz WBAN,” Biosensors 2023, 13(1), 73; https://doi.org/10.3390/bios13010073” and literature 2:“Design and optimization of an ultrathin and broadband polarization-insensitive fractal FSS using the improved bacteria foraging optimization algorithm and curve fitting,” Nanomaterials 2023, 13(1), 191; https://doi.org/10.3390/nano13010191”and this manuscript. And explain whether the new methods of the above two literatures can be applied to this manuscript.
Author Response
We sincerely thank the reviewer for the valuable and insightful comments/suggestions to improve the quality of this manuscript. Please find attached the details of our response to the reviewer’s comments. For convenience, all changes are marked in red in the revised manuscript.

Reviewer 2 Report
This paper investaigates Water Surface Targets Detection Based on the Fusion of Vision and LiDAR. I recommend the following:
1. The statistical significance of the detection and positioning should be discussed at what confidence level this results are valid.
2. The assumption of threshold and weight coeffcients makes this method addaptive method. This should be emphasized in the paper.
3. In abstract and conclusion, the percentage of improvement of the proposed fusion method over the classical method should be mentioned.
Author Response

(The authors gave the same response as above.)

Round 2
Reviewer 1 Report
This paper can accept to publish.